# *Chrysanthemum morifolium* Extract Ameliorates Doxorubicin-Induced Cardiotoxicity by Decreasing Apoptosis

**DOI:** 10.3390/cancers14030683

**Published:** 2022-01-28

**Authors:** Masaya Ono, Yoichi Sunagawa, Saho Mochizuki, Takahiro Katagiri, Hidemichi Takai, Sonoka Iwashimizu, Kyoko Inai, Masafumi Funamoto, Kana Shimizu, Satoshi Shimizu, Yasufumi Katanasaka, Maki Komiyama, Philip Hawke, Hideo Hara, Yoshiki Arakawa, Kiyoshi Mori, Akira Asai, Koji Hasegawa, Tatsuya Morimoto

**Affiliations:** 1Division of Molecular Medicine, School of Pharmaceutical Sciences, University of Shizuoka, Shizuoka 422-8526, Japan; m17026@u-shizuoka-ken.ac.jp (M.O.); y.sunagawa@u-shizuoka-ken.ac.jp (Y.S.); m15110@u-shizuoka-ken.ac.jp (S.M.); m17028@u-shizuoka-ken.ac.jp (T.K.); m17067@u-shizuoka-ken.ac.jp (H.T.); m18114@u-shizuoka-ken.ac.jp (S.I.); m19006@u-shizuoka-ken.ac.jp (K.I.); funamoto@u-shizuoka-ken.ac.jp (M.F.); s18804@u-shizuoka-ken.ac.jp (K.S.); s18410@u-shizuoka-ken.ac.jp (S.S.); katana@u-shizuoka-ken.ac.jp (Y.K.); koj@kuhp.kyoto-u.ac.jp (K.H.); 2Division of Translational Research, Clinical Research Institute, Kyoto Medical Center, National Hospital Organization, Kyoto 612-8555, Japan; nikonikomakirin@yahoo.co.jp; 3Shizuoka General Hospital, Shizuoka 420-8527, Japan; mori@u-shizuoka-ken.ac.jp; 4Laboratory of Scientific English, School of Pharmaceutical Sciences, University of Shizuoka, Shizuoka 422-8526, Japan; hawke@u-shizuoka-ken.ac.jp; 5UNIAL Co., Ltd., Tokyo 173-0005, Japan; h-hara@unial.co.jp; 6Department of Neurosurgery, Kyoto University Graduate of Medicine, Kyoto 606-8507, Japan; yarakawa@kuhp.kyoto-u.ac.jp; 7Graduate School of Public Health, Shizuoka Graduate University of Public Health, Shizuoka 420-0881, Japan; 8Department of Molecular and Clinical Pharmacology, School of Pharmaceutical Sciences, University of Shizuoka, Shizuoka 422-8526, Japan; 9Center for Drug Discovery, Graduate School of Pharmaceutical Sciences, University of Shizuoka, Shizuoka 422-8526, Japan; aasai@u-shizuoka-ken.ac.jp

**Keywords:** *Chrysanthemum morifolium*, doxorubicin, cardiomyopathy, apoptosis, p53, systolic dysfunction

## Abstract

**Simple Summary:**

The anticancer drug doxorubicin is widely used for the treatment of malignant tumors, including colon, breast, and ovary cancers. However, prolonged use of doxorubicin causes heart damage, ranging from changes in the structure and function of heart cells to heart failure, the condition in which the heart does not pump enough blood. As this problem affects the quality of life and survival of cancer patients, solutions to it are urgently needed. This study demonstrates that *Chrysanthemum morifolium* extract, an extract of the purple chrysanthemum flower, reduced the heart damage caused by doxorubicin by suppressing cell death in heart cells and heart failure in an animal model. As *Chrysanthemum morifolium* has been eaten since ancient times, the extract from this functional food is likely to be safe in clinical application, potentially allowing patients to receive the well-established anti-cancer benefits of doxorubicin without the side effect of heart damage.

**Abstract:**

It is well known that the anthracycline anticancer drug doxorubicin (DOX) induces cardiotoxicity. Recently, *Chrysanthemum morifolium* extract (CME), an extract of the purple chrysanthemum flower, has been reported to possess various physiological activities such as antioxidant and anti-inflammatory effects. However, its effect on DOX-induced cardiotoxicity is still unknown. An 3-(4,5-Dimethylthiazol-2-yl)-2,5-Diphenyltetrazolium Bromide (MTT)assay revealed that 1 mg/mL of CME reduced DOX-induced cytotoxicity in H9C2 cells but not in MDA-MB-231 cells. A TUNEL assay indicated that CME treatment improved DOX-induced apoptosis in H9C2 cells. Moreover, DOX-induced increases in the expression levels of p53, phosphorylated p53, and cleaved caspase-3,9 were significantly suppressed by CME treatment. Next, we investigated the effect of CME in vivo. The results showed that CME treatment substantially reversed the DOX-induced decrease in survival rate. Echocardiography indicated that CME treatment also reduced DOX-induced left ventricular systolic dysfunction, and a TUNEL assay showed that CME treatment also suppressed apoptosis in the mouse heart. These results reveal that CME treatment ameliorated DOX-induced cardiotoxicity by suppressing apoptosis. Further study is needed to clarify the effect of CME on DOX-induced heart failure in humans.

## 1. Introduction

In recent years, the prognosis of patients with malignant neoplasm has been improving due to advances in surgical treatment, radiation therapy, and chemotherapy with anticancer drugs [1,2]. However, the long-term survival of patients has increased the amount of cardiotoxicity caused by anticancer drugs, which had not been previously observed [3,4]. As cardiotoxicity has a major impact on the prognosis and quality of life of cancer patients, reducing it is an urgent issue, both clinically and economically [5].

Cardiotoxicity caused by anticancer drugs is classified into two types according to clinical characteristics [6]. Type I anti-tumor agents, represented by anthracyclines, induce dose-dependent myocardial disorders with irreversible histochemical changes. Type II anti-tumor agents are dose-independent drugs that induce myocardial dysfunction with reversible histochemical changes. Among the anthracyclines, doxorubicin (DOX) is indicated for various cancers, including malignant lymphoma, lung cancer, gastrointestinal cancer, breast cancer, and osteosarcoma. It is highly effective and is considered an essential treatment for these cancers. However, the risk of DOX-induced cardiotoxicity increases cumulatively and dose dependently, and studies of cancer patients treated with DOX reported that from 3 to 26% of patients developed heart failure [7,8,9]. In order to avoid this DOX-induced heart failure, the drug is no longer used for cancer patients with a history of its use or for patients with cardiac disfunction [10], thus depriving these patients of the therapeutic benefits of DOX.

DOX induces cardiotoxicity by generating reactive oxygen species (ROS) through the direct or indirect chelation of free Fe2+ and by breaking double stranded DNA through interaction with topoisomerase II β (TOP2β). This DNA damage then upregulates the tumor suppressor gene p53, which activates DNA repair proteins. These proteins also suppress the expression of genes involved in mitochondrial biosynthesis and antioxidant activity, leading to mitochondrial deficiency [11,12,13,14]. In addition, the activation of p53 enhances the transcription of Bax, a Bcl-2 family protein, which acts on mitochondria to promote the release of cytochrome c. The released cytochrome c activates caspase 9, which activates caspase 3, leading to the induction of apoptosis in cardiomyocytes and eventually to the development of heart failure [15,16].

A wide variety of compounds found in foods and plants have been reported to exhibit various physiological functions, including antioxidant [17] and antitumor [18] activity and the suppression of heart failure [19,20,21,22]. Several naturally occurring compounds have been reported to suppress DOX-induced cardiotoxicity in vitro by inhibiting apoptosis; however, there are few reports of animal and clinical studies on these compounds. *Chrysanthemum morifolium* extract (CME) is an extract from the purple chrysanthemum flower containing large amounts of luteolin and chlorogenic acid. It has been reported to have various physiological effects, such as anti-inflammatory and free radical scavenging activity [23,24,25,26]. However, the effect of CME on DOX-induced cardiotoxicity has not been reported. In this study, we investigated the effects of CME on DOX-treated H9C2 rat cardiomyoblast cells and on DOX-induced cardiomyopathy in mice.

## 2. Materials and Methods

### 2.1. Materials

CME was obtained from UNIAL Co., Ltd. (Tokyo, Japan). *Chrysanthemum morifolium* was extracted with hot water, then drying it into a powder. This extract is a standardized product including chlorogenic acid (420 mg/100 g), delphinidin (690 mg/100 g), luteolin (32 mg/100 g), and total polyphenol (4.75 g/100 g) derived from *Chrysanthemum morifolium* Ramat. CME is available on UNIAL’s website (http://www.unial.info/materials/beauty.html (accessed on 28. January 2022). Doxorubicin hydrochloride was purchased from MedChemExpress (Monmouth Junction, NJ, USA) and stored at −20 °C until use.

### 2.2. Cell Culture and CME Treatment

Primary cultured neonatal rat cardiomyocytes were isolated from 1- to 2-day-old SD rats as described previously [20,27]. H9C2 cells, MDA-MB-231 cells, H1299 cells, and HT29 cells. were purchased from American Type Culture Collection (Manassas, VA, USA). These cells were cultured in Dulbecco’s Modified Eagle’s Medium (Nacalai Tesque, Kyoto, Japan) with fetal bovine serum (FBS) and 1% penicillin-streptomycin-glutamine (Invitrogen, Carlsbad, CA, USA) in a humidified incubator at 37 °C with 5% CO_2_. These cells were treated with 0.3 or 1 mg/mL CME for 2 h, followed by stimulation with 1 µM DOX. H9C2 cells were incubated for 24 h for the MTT assay and 12 h for protein extraction. Primary cultured cardiomyocytes were incubated for 24h for the MTT assay. MDA-MB-231 cells were incubated for 48 h for the MTT assay and 24 h for protein extraction. H1299 cells and HT29 cells were incubated for 48 h for the MTT assay.

### 2.3. MTT Cell Viability Assay

H9C2 cells, primary cultured cardiomyocytes, MDA-MB-231 cells, H1299 cells, and HT29 cells were washed with serum free medium and added to Counting Kit-8 medium (Dojindo, Kumamoto, Japan) for 1 h. After incubation, measurement of absorbance was performed using a Wallac 1420 Arvo Sx multilabel counter (Perkin Elmer, Waltham, MA, USA). The relative percentage of cell survival was calculated by dividing the absorbance of the treated cells by that of the control in each experiment.

### 2.4. Animal Experiments

C57BL/6 J male mice were purchased from Japan SLC Inc. (Shizuoka, Japan). The mice were randomly assigned to three groups: vehicle (1% gum Arabic, *n* = 10), 20 mg/kg DOX (*n* = 9), and 20 mg/kg DOX + 400 mg/kg CME (*n* = 10). CME was administrated to the mice orally by gastric gavage once a day for 14 days beginning 2 days before DOX injection. To determine survival, mice mortality was monitored for 12 days after DOX injection. 

### 2.5. Western Blotting

Protein extracts and nuclear extracts were obtained from H9C2 cells and MDA-MB-231 cells. Western blotting was performed as previously described [27,28,29]. For western blotting, anti-cleaved caspase-3 antibody, anti-cleaved caspase-9 antibody (Abcam, Cambridge, UK), anti-p53 antibody, phospho-p53 (Ser15) antibody (Cell Signaling Technology, Danvers, MA, USA), and anti α-tubulin monoclonal antibody (Fujifilm Wako Pure Chemical Corporation, Osaka, Japan) were used as primary antibodies, and goat anti-rabbit IgG–HRP (MBL, Aichi, Japan) and sheep anti-mouse IgG-HRP (GE Healthcare, Chicago, IL, USA) were used as secondary antibodies.

### 2.6. TUNEL Staining

The terminal deoxynucleotidyl transferase-mediated dUTP nick-end labeling (TUNEL assay was performed in accordance with the manufacturer’s protocol (Roche, Basel, Switzerland) as previously described [30,31]. In brief, H9C2 cells were fixed in 10% formalin. Following permeabilization with 0.3% Triton X-100 in PBS, cells were incubated with TUNEL reaction solution (In Situ Cell Death Detection Kit, TMR red, Roche). The mice were euthanized, and their hearts were isolated and cut into two transverse slices at the mid-level of the papillary muscles. The samples were fixed with Optimal Cutting Temperature Compound (Sakura Finetek Japan Co., Ltd., Osaka, Japan) and then in 10% formalin. They were quenched with PBS containing 100 mM Tris-HCl and 0.1% Triton X-100, and were then permeabilized with 0.3% Triton X-100 in PBS. After permeabilization, they were incubated with TUNEL reaction solution. Following the application of a TrueView Autofluorescence Quenching Kit (Vector Laboratories, Burlingame, CA, USA) to reduce cellular autofluorescence signal in cardiac tissue, the samples were incubated at room temperature. Nuclear staining was then performed with 1 µg/mL Hoechest 33258 (Dojindo, Kumamoto, Japan) for 2 h at 4 °C. Fluorescence was observed with a fluorescence microscope (BZ-X810, Keyence, Osaka, Japan). To determine the percentage of apoptotic cells, TUNEL-positive nuclei and TUNEL-negative cells were counted using Image J software, version 1.51 (U.S. National Institutes of Health, Bethesda, MD, USA). Samples from at least three independent experiments were scored blindly.

### 2.7. Echocardiography

Echocardiography was performed using a 10–12 MHz probe and a Sonos 5500 Ultrasound System (Philips, Amsterdam, The Netherlands) as described previously [27,32,33]. Left ventricular internal diameter end-diastole (LVIDd), left ventricular internal diameter end-systole (LVIDs), and left ventricular posterior wall thickness (LVPWT) were obtained from M-mode recordings. Fractional shortening (FS) was calculated as (LVIDd  −  LVIDs)/LVIDd × 100 (%). 

### 2.8. Statistics

Values are shown as the mean ± SEM from at least three independent experiments. Statistical comparisons were performed using ANOVA with the Tukey–Kramer test. Survival rate was analyzed by log-rank test. A *p* value of < 0.05 was considered statistically significant.

## 3. Results

### 3.1. CME Inhibited DOX-Induced Cytotoxicity in H9C2 Cells and Primary Cultured Cardiomyocytes

The effect of CME on DOX-induced cytotoxicity was examined using an MTT assay with H9C2 cells and primary cultured cardiomyocytes (Figure 1A,B). DOX reduced the viability of these cells to 29% and 28%, respectively, but 1 mg/mL CME inhibited DOX-induced cell cytotoxicity, increasing cell viability to 75% and 79%, respectively. An MTT assay also revealed that 1 mg/mL CME did not induce cytotoxicity in H9C2 cells (Figure 1C). Next, to determine whether CME reduces the anti-tumor activity of DOX, MDA-MB-231 human breast cancer cells, H1299 cells human non-small cell lung carcinoma cells and HT29 human colon cancer cells were treated with CME and DOX. Cell viability was decreased by DOX, but 1 mg/mL CME did not affect viability (Figure 1D–F). These results indicate that CME inhibited the cytotoxicity induced by DOX without reducing its anti-tumor activity.

### 3.2. CME Inhibited DOX-Induced Apoptosis in H9C2 Cells

To investigate the effects of CME on DOX-induced apoptosis in vitro, TUNEL staining was performed on H9C2 cells. The results showed that DOX-induced apoptosis was inhibited by CME (Figure 2).

### 3.3. CME Inhibited DOX-Induced Upregulation of p53, p-p53, Cleaved Caspase-3, and Cleaved Caspase-9

First, to investigate whether CME affects the expression levels of p53, phosphorylated p53 (p-p53), cleaved caspase-3, and cleaved caspase-9, H9C2 cells were pretreated with 1 mg/mL CME for 2 h and then incubated with 1 µM DOX for 12 h. One mg/mL CME significantly inhibited DOX-induced upregulation of p53 and p-p53 (Figure 3A–C). Moreover, 1 mg/mL CME inhibited DOX-induced increases in cleaved caspase-3 and -9 activity (Figure 3D–F). Next, to determine whether CME inhibits DOX-induced upregulation of p53, MDA-MB-231 cells were pretreated with 1 mg/mL CME for 2 h and then incubated with 1 µM DOX for 24 h. Western blotting demonstrated that 1 mg/mL CME did not affect DOX-induced increases in p53 upregulation (Figure 3G). These results revealed that CME inhibited DOX-induced apoptosis by inhibiting p53 and cleaved casapase-3 activity.

### 3.4. CME Improved a DOX-Induced Decrease in Survival Rate

DOX-induced cardiomyopathy model mice were used to investigate the effects of CME on DOX-induced cardiotoxicity. Eight-week-old C57BL/6J male mice were randomly assigned treatment with vehicle, DOX, or 400 mg/kg CME + DOX. CME or solvent was administrated to the mice orally for 15 days beginning 2 days before an intraperitoneal injection of DOX. As shown in Figure 4, a DOX-induced decrease in survival rate was improved by CME.

### 3.5. CME Improved DOX-Induced Cardiac Dysfunction 

Cardiac function was assessed by echocardiography 7 days after intraperitoneal injection of DOX (Figure 5A). The results indicated that FS and EF were decreased by DOX, and that CME suppressed these changes (Figure 5B,C). Body weight was decreased in the DOX and DOX + CME groups compared to the vehicle group (Figure 5D). CME improved a DOX-indued decrease in the ratio of heart weight to tibia length (HW/TL) (Figure 5E). These results indicate that CME improved DOX-induced cardiac disfunction in mice.

### 3.6. CME Suppressed DOX-Induced Cardiac Apoptosis in Mice

To determine the protective effect of CME against DOX-induced cardiac damage, a histological analysis was carried out. Representative myocardial cross-sectional images stained with hemotoxin/eosin and picrosirius red are shown in Appendix A. There were no significant differences in cardiac morphology, collagen, or fibrosis in the heart tissue between the DOX-treated group and Control group. TUNEL staining was performed to investigate the effect of CME on DOX-induced cardiac apoptosis in vivo (Figure 6A). Apoptosis in the heart was increased by DOX, and this increase was suppressed by CME (Figure 6B). These results reveal that CME improved DOX-induced cardiac dysfunction by inhibiting DOX-induced apoptosis.

## 4. Discussion

This study found that CME suppressed DOX-induced cytotoxicity in H9C2 cells and primary cultured myocytes without inhibiting the anti-tumor effect of DOX in MDA-MB-231 cells, H1299 cells and HT29 cells. This suggests that CME suppresses DOX-induced cardiotoxicity without inhibiting the anti-tumor effect of DOX.

CME also suppressed in vitro the activation of caspase-3 and -9, which regulate DOX-induced apoptosis, as well as that of the tumor suppressor gene p53, which is upstream of these caspases in H9C2 cells. In contrast, CME didn’t affect DOX-induced upregulation of p53 in MDA-MB-231. DOX is known to cause DNA damage by the interaction of three factors: ROS production and a subsequent increase in oxidative stress, mitochondrial metabolic dysfunction, and the interaction of damaged DNA with TOP2β. This DNA damage then activates p53 [34,35,36]. CME has been reported to possess mainly anti-inflammatory and anti-tumor activities [23,25,37]. The results of the present study show that CME inhibited the activation of p53. This suggests that CME inhibits the activation of p53 by affecting oxidative stress, mitochondrial dysfunction, and the interaction of damaged DNA with TOP2β in cardiomyocytes.

To investigate whether CME inhibits DOX-induced cardiotoxicity in vivo, we used DOX-induced heart failure model mice. Echocardiography showed that CME suppressed cardiac dysfunction due to DOX, and TUNEL staining revealed that CME suppressed DOX-induced apoptosis in vivo. Wencker, et al. reported that low levels of myocyte apoptosis (23 myocytes per 10^5^ nuclei) were sufficient to cause a lethal, dilated cardiomyopathy of in mice with cardiac specific expression of caspase-8 fusion protein [38]. Our study showed that 26 myocytes per 10^5^ (2.6 × 10^−2^%) nuclei were detected at one week after DOX administration, indicating this ratio can be considered sufficient to have induced cardiac dysfunction in vivo. These results indicate that CME improves DOX-induced cardiotoxicity by suppressing apoptosis, which is a major cause of the progression of heart failure.

Luteolin and chlorogenic acid, which are the main components of CME, are known to have antioxidant and anti-inflammatory effects. As oxidative stress and inflammatory response cause DNA damage, activate p53, and induce cardiac apoptosis in DOX treated mice, the antioxidant and anti-inflammatory effects of CME may suppress DOX-induced cardiotoxicity [23,25,37]. Studies using isoproterenol-induced cardiomyopathy model mice have shown that luteolin-7-diglucuronide, a glycoside of luteolin, prevents cardiomyopathy by suppressing the expression of the gene encoding the subunit of NADPH oxidase that is responsible for ROS production. This suggests that luteolin in CME may also suppress oxidative stress and have a cardioprotective effect [39,40]. Several studies have investigated the pharmacological activity of CME. In a hypertension rat model, CME reduced blood pressure and improved cardiac hypertrophy [41]. This suggests that CME may prevent not only antitumor drug-induced cardiotoxicity but also hypertension-induced heart failure. In addition, CME has been shown to possess anti-diabetic, antitumor, and antiviral effects [42,43,44,45,46]. These findings suggest that CME may have a preventive effect on various diseases due to its anti-inflammatory and antioxidant effects. 

A previous study evaluating the safety of CME showed that no toxicity was observed even after daily oral administration of 1280 mg/kg to SD rats for 26 weeks [47]. As *Chrysanthemum morifolium* has been eaten since ancient times, it is likely to be safe. Nevertheless, further clinical safety studies on CME should still be carried out.

This study has several limitations. CME is known to contain large amounts of delphinidin, luteolin, and chlorogenic acid; however, in this study we were unable to identify which compounds contributed to the beneficial effect of CME on DOX-induced cardiotoxicity. Second, to determine the effect of CME, this study focused only on the acute cardiotoxicity induced by a single high dose of DOX. Therefore, based on this study, it is not possible to predict the protective effect of CME on chronic cardiomyopathy induced by low doses of DOX. Recently, human pluripotent stem cell-derived cardiomyocytes and endothelial cells have emerged as useful tools for analyzing cardiotoxicity in physiologically relevant human cells [48]. As this study confirmed the safety of CME only in primary cultured cardiomyocytes and H9C2 cells, future studies are needed to clarify the safety of the compound in human pluripotent stem cell-derived cardiomyocytes and endothelial cells in order to predict potential adverse effects of DOX in the clinical setting.

## 5. Conclusions

In summary, this study demonstrates that CME suppresses DOX-induced cytotoxicity, apoptosis, and cardiac dysfunction without inhibiting the antitumor activity of DOX. More detailed studies of the mechanism of CME on DOX-induced cardiotoxicity may lead to the development of a novel therapy that can bring the well-established anti-cancer benefits of DOX to patients without the side effect of heart failure.

## Figures and Tables

**Figure 1 cancers-14-00683-f001:**
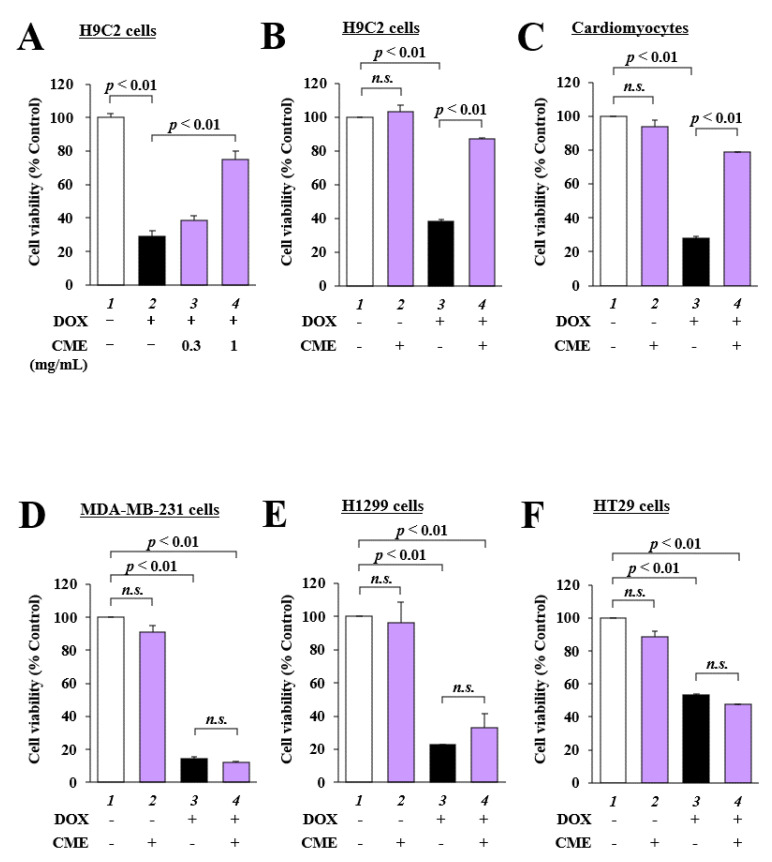
DOX-induced cytotoxicity was inhibited by CME in H9C2 cells and primary cultured cardiomyocytes. (**A**) H9C2 cells were pretreated with 0.3 or 1 mg/mL CME. After 2 h, the cells were treated with 1 µM DOX for 24 h. Cell viability was investigated by MTT assay. (**B**,**C**) H9C2 cells and primary cultured cardiomyocytes were pretreated with 1 mg/mL CME for 2 h and then treated with 1 µM DOX for 24 h. (**D**–**F**) MDA-MB-231 cells (**D**), H1299 cells (**E**), and HT29 cells (**F**) were pretreated with 1 mg/mL CME. These cells were stimulated with 1 µM DOX for 48 h. Cell viability was measured by MTT assay. Values are presented as the mean ± SEM of three individual experiments.

**Figure 2 cancers-14-00683-f002:**
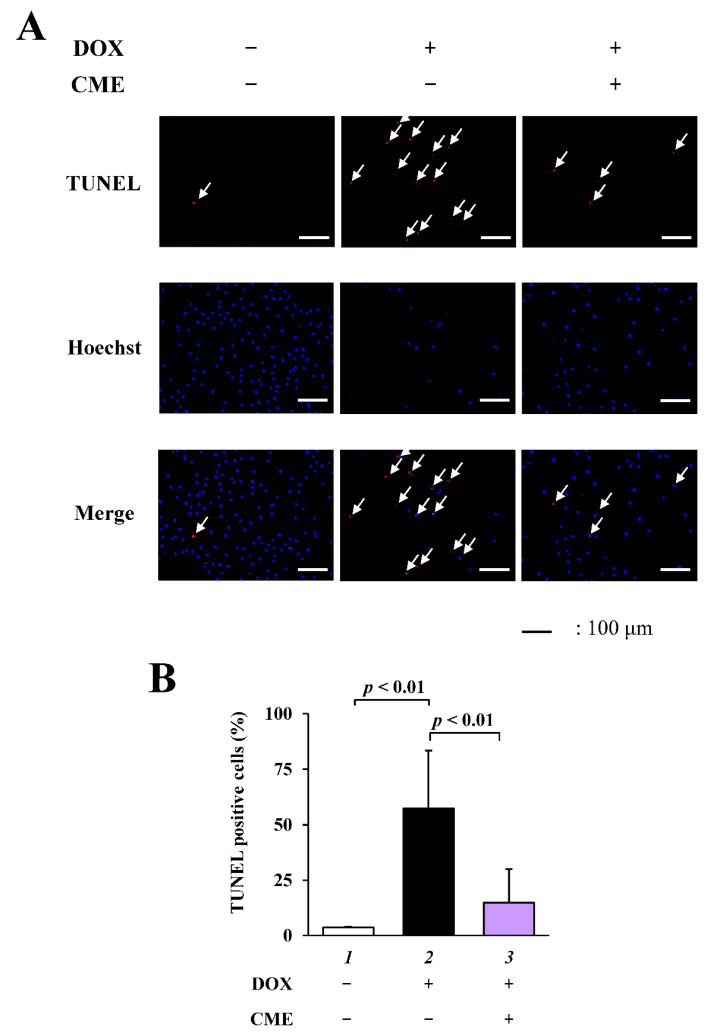
DOX-induced apoptosis was inhibited by CME in H9C2 cells. H9C2 cells were pretreated with 1 mg/mL CME. After 2 h, the cells were treated with 1 µM DOX. Twenty-four hours after treatment, TUNEL assay and nuclear staining were performed. (**A**) Representative images of TUNEL staining and nuclear staining with 1 µg/mL Hoechest 33258 of H9C2 cells. Arrows show TUNEL positive cells. (**B**) TUNEL positive ratio is defined as the number of TUNEL positive cells divided by the total number of cells. Values are presented as the mean ± SEM of three individual experiments.

**Figure 3 cancers-14-00683-f003:**
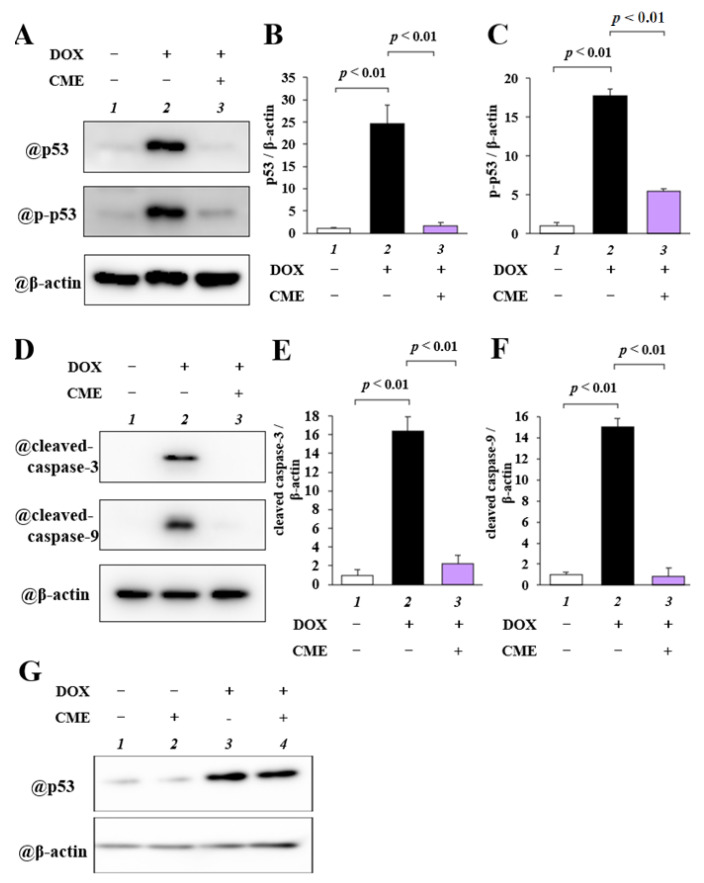
DOX-induced protein expression of cellular apoptosis markers was inhibited by CME. H9C2 cells were pretreated with 1mg/mL CME for 2 h, and then cardiac cytotoxicity was induced with 1 µM DOX for 12 h. MDA-MB-231 cells were pretreated with 1mg/mL CME for 2 h, and then cardiac cytotoxicity was induced with 1 µM DOX for 24 h. (**A**,**D**) Representative images of WB in H9C2 cells. (**B**,**C**,**E**,**F**) Expression levels of p53 (**B**), phosph-p53 (**C**), cleaved caspase-3 (**E**), and cleaved caspase-9 (**F**) were calculated as ratios relative to β-actin. (**G**) Representative images of WB in MDA-MB-231 cells. Values are presented as the mean ± SEM of three individual experiments.

**Figure 4 cancers-14-00683-f004:**
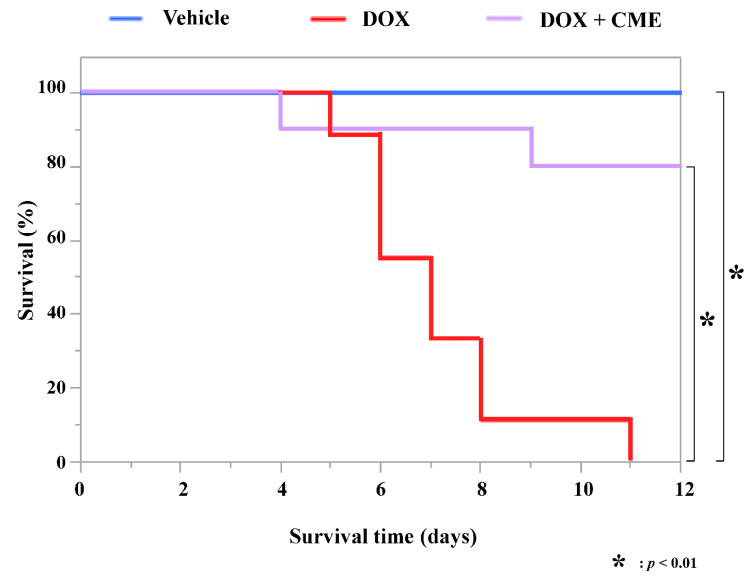
Survival rate that had been decreased by DOX was improved by CME. Survival rate was determined for 12 days after intraperitoneal injection of 20 mg/kg DOX. Blue, vehicle (*n* = 10); red, DOX (*n* = 9); purple, 400 mg/kg CME + DOX (*n* = 10). * *p* < 0.01.

**Figure 5 cancers-14-00683-f005:**
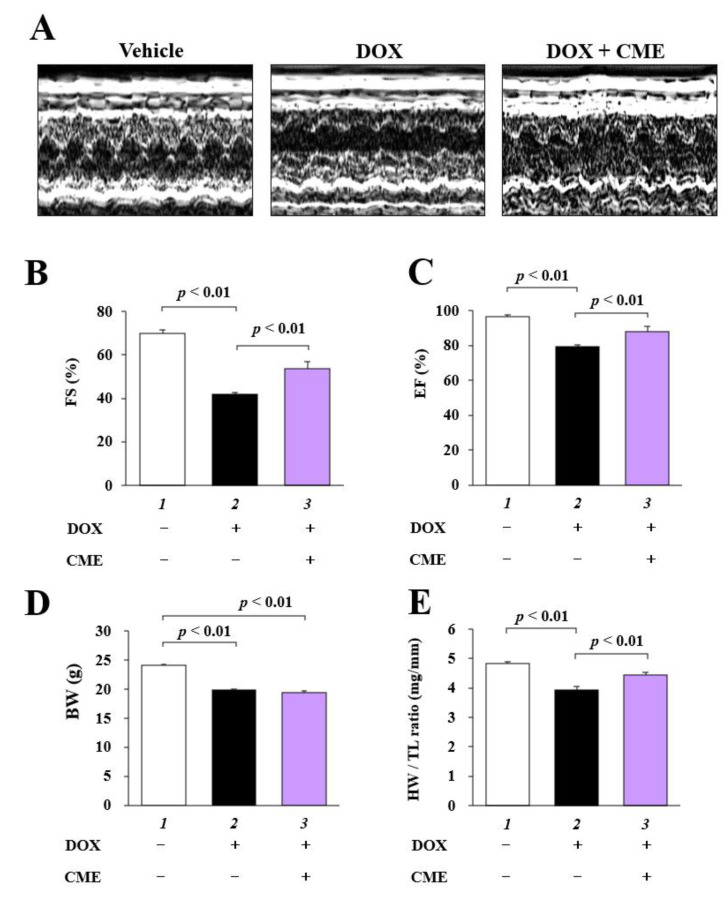
DOX-induced cardiac dysfunction was inhibited by CME. (**A**) Representative photographs of M-mode images. (**B**) FS, fractional shortening. (**C**) EF, ejection fraction. (**D**) BW, body weight. (**E**) HW, heart weight; TL, tibia length. Values are presented as the mean ± SEM of 10 individual measurements.

**Figure 6 cancers-14-00683-f006:**
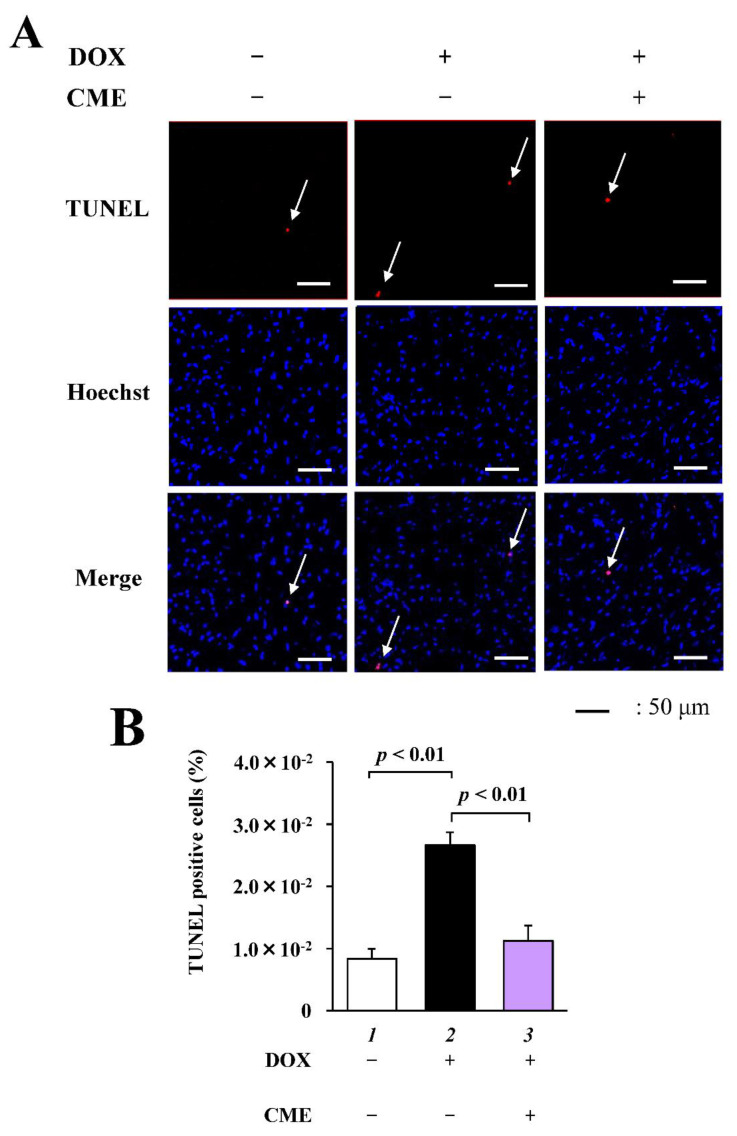
DOXinduced apoptosis in mouse heart was inhibited by CME. (**A**) Representative images of TUNEL staining and nuclear staining with 1 µg/mL Hoechest 33258 of mouse heart. Arrows show TUNEL positive cells. (**B)** TUNEL positive ratio is defined as the number of TUNEL positive cells divided by the total number of cells. Values are presented as the mean ± SEM of three individual experiments.

## Data Availability

The data presented in this study are available on request from the corresponding author.

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
