# Peer review of "Chrysanthemum morifolium Extract Ameliorates Doxorubicin-Induced Cardiotoxicity by Decreasing Apoptosis"

_cancers, 2022, doi:10.3390/cancers14030683_

Round 1

Reviewer 1 Report

This manuscript of Morimoto et al., deals with the biological studies to investigate the potential of Chrysanthemum morifolium extract (CME) to reduce the cardiotoxicity of the doxorubicin, clinically used anticancer agent. To begin with, the manuscript clearly indicates their purpose of research by providing necessary backgrounds for the readers to understand the problem it wants to address and suggest the solutions based on the outcomes from their literature search. Thus, the readers were able to clearly understand the research purpose. As a reviewer, I felt interested in the concept of this paper.

Several biological studies were done to investigate the biological activity of the CME. One of them is MTT cell viability assay. In this manuscript, this assay was used to evaluate the impact of CME on the toxicity of the doxorubicin. This is a good approach and the choice of MDA-MB-231 is very appropriate, since doxorubicin is used to treat breast cancer in a combination.; however, the manuscript only involved single cancer cell type to investigate the anticancer activity. It is recommended to add another cell line, e.g. ovarian cancer cell line to verify that CME effect is cancer cell line-independent.

The authours should add more experimental details in figure legends including concentration of CME, treatment time, concentration of DOXO, concentration of Hoechst, number of mice in the group, etc. Most of these details are available from experimental data, however, it will facilitate readers’ understanding of the figures if the details are added in the legends. The authours should also describe the details of Hoechst staining and explain what is SFM in Experimental section.

As for the western blotting, the overall results look quite convincing. Not only the blotting shapes look decent, but also, the quantifications of proteins in different treatments indicate the influence of the CME on the doxorubicin therapy. However, it is also important to perform Western Blotting in one cancer cell line to correlate the effects with the MTT data.

Overall, I like the manuscript very much, I think that the manuscript is a good fit for the journal and will be well cited.

I have one major comment that must be addressed. The authours mention that CME is commercially available. In that case they should give the catalogue number, purity and other details. If it is not available online, the authours should provide the data about the composition of the CME – HPLC, types of fractions, purity, etc. Without this data, it is close to impossible to reproduce these results.

Author Response

Reviewer 1

Comments and Suggestions for Authors

This manuscript of Morimoto et al., deals with the biological studies to investigate the potential of Chrysanthemum morifolium extract (CME) to reduce the cardiotoxicity of the doxorubicin, clinically used anticancer agent. To begin with, the manuscript clearly indicates their purpose of research by providing necessary backgrounds for the readers to understand the problem it wants to address and suggest the solutions based on the outcomes from their literature search. Thus, the readers were able to clearly understand the research purpose. As a reviewer, I felt interested in the concept of this paper.

Comment 1-1:

Several biological studies were done to investigate the biological activity of the CME. One of them is MTT cell viability assay. In this manuscript, this assay was used to evaluate the impact of CME on the toxicity of the doxorubicin. This is a good approach and the choice of MDA-MB-231 is very appropriate, since doxorubicin is used to treat breast cancer in a combination.; however, the manuscript only involved single cancer cell type to investigate the anticancer activity. It is recommended to add another cell line, e.g. ovarian cancer cell line to verify that CME effect is cancer cell line-independent.

Response 1-1:

Thank you very much for your valuable suggestion. In response to it, we have carried out additional experiments and found that CME did not reduce the anti-tumor activity of DOX in H1299 human non-small cell lung carcinoma cells or in HT29 human colon cancer cells. These results suggest that CME inhibits the cardiac cytotoxicity induced by DOX without reducing its anti-tumor activity against various cancers. We have revised the Method, Results and Discussion sections.

Line 98 on Page 4, Materials and Methods

“Primary cultured neonatal rat cardiomyocytes were isolated from 1- to 2-day-old SD rats as described previously [20, 49]. H9C2 cells, MDA-MB-231 cells, and H1299 cells were purchased from American Type Culture Collection (USA). HT29 cells were kindly provided by Prof. Akira Asai (University of Shizuoka, Japan). These cells were cultured in Dulbecco’s Modified Eagle’s Medium (Nacalai Tesque, Japan) with fetal bovine serum (FBS) and 1% penicillin-streptomycin-glutamine (Invitrogen, USA) in a humidified incubator at 37°C with 5% CO2. These cells were treated with 0.3 or 1 mg/mL CME for 2 h, followed by stimulation with 1 µM DOX. H9C2 cells were incubated for 24 h for the MTT assay and 12 h for protein extraction. Primary cultured cardiomyocytes were incubated for 24h for the MTT assay. MDA-MB-231 cells were incubated for 48 h for the MTT assay and 24 h for protein extraction. H1299 cells and HT29 cells were incubated for 48 h for the MTT assay.”

Line 107 on Page 4, Materials and Methods

“H9C2 cells, primary cultured cardiomyocytes, MDA-MB-231 cells, H1299 cells, and HT29 cells were washed with serum free medium and added to Counting Kit-8 medium (Dojindo, Japan) for 1 h. After incubation, measurement of absorbance was performed using a Wallac 1420 Arvo Sx multilabel counter (Perkin Elmer, USA). The relative percentage of cell survival was calculated by dividing the absorbance of the treated cells by that of the control in each experiment.”

Line 177 on Page 5, Results

3.1. CME inhibited DOX-induced cytotoxicity in H9C2 cells and primary cultured cardiomyocytes.

The effect of CME on DOX-induced cytotoxicity was examined using an MTT assay with H9C2 cells and primary cultured cardiomyocytes (Fig. 1A, B). DOX reduced the viability of these cells to 29% and 28%, respectively, but 1 mg/mL CME inhibited DOX-induced cell cytotoxicity, increasing cell viability to 75% and 79%, respectively. An MTT assay also revealed that 1 mg/mL CME did not induce cytotoxicity in H9C2 cells (Fig. 1C). Next, to determine whether CME reduces the anti-tumor activity of DOX, MDA-MB-231 human breast cancer cells, H1299 cells human non-small cell lung carcinoma cells and HT29 human colon cancer cells were treated with CME and DOX. Cell viability was decreased by DOX, but 1 mg/mL CME did not affect viability (Fig. 1D-F). These results indicate that CME inhibited the cytotoxicity induced by DOX without reducing its antitumor activity.”

Line 172 on Page 5, Figure legends

Figure 1. DOX-induced cytotoxicity was inhibited by CME in H9C2 cells and primary cultured cardiomyocytes.

(A) H9C2 cells were pretreated with 0.3 or 1 mg/mL CME. After 2 h, the cells were treated with 1 µM DOX for 24 h. Cell viability was investigated by MTT assay. (B, C) H9C2 cells and primary cultured cardiomyocytes were pretreated with 1 mg/mL CME for 2 h and then treated with 1 µM DOX for 24 h. (D-F) MDA-MB-231 cells (D), H1299 cells (E), and HT29 cells (F) were pretreated with 1 mg/mL CME. These cells were stimulated with 1 µM DOX for 48 h. Cell viability was measured by MTT assay. Values are presented as the mean ± SEM of three individual experiments.”

Line 250 on Page 10, Discussion

“This study found that CME suppressed DOX-induced cytotoxicity in H9C2 cells and primary cultured myocytes without inhibiting the anti-tumor effect of DOX in MDA-MB-231 cells, H1299 cells and HT29 cells. This suggests that CME suppresses DOX-induced cardiotoxicity without inhibiting the anti-tumor effect of DOX”

Line 287 on Page 9, Institutional Review Board Statement

“Institutional Review Board Statement: All animal experiments complied with the guidelines on animal experiments of the University of Shizuoka and the National Hospital Organization Kyoto Medical Center and were performed in accordance with protocols approved by the University of Shizuoka Ethics Committee (numbers 176278 and 186353) and the National Hospital Organization Kyoto Medical Center Ethics Committee (number 30-29-1).”

Comment 1-2:

The authours should add more experimental details in figure legends including concentration of CME, treatment time, concentration of DOXO, concentration of Hoechst, number of mice in the group, etc. Most of these details are available from experimental data, however, it will facilitate readers’ understanding of the figures if the details are added in the legends. The authours should also describe the details of Hoechst staining and explain what is SFM in Experimental section.

Response 1-2:

                 Thank you for your helpful advice. According to your suggestion, we have modified the Method section and Figure legends.

Line 107 on Page 4, Method

“H9C2 cells, primary cultured cardiomyocytes, MDA-MB-231 cells, H1299 cells and HT29 cells were washed with serum free medium and added to Counting Kit-8 medium (Dojindo, Japan) for 1 h. After incubation, measurement of absorbance was performed using a Wallac 1420 Arvo Sx multilabel counter (Perkin Elmer, USA). The relative percentage of cell survival was calculated by dividing the absorbance of the treated cells by that of the control in each experiment “

Line 130 on Page 4, Method

“The TUNEL (terminal deoxynucleotidyl transferase-mediated dUTP nick-end labeling) assay was performed in accordance with the manufacturer’s protocol (Roche, Switzerland) as previously described [30, 31]. In brief, H9C2 cells were fixed in 10% formalin. Following permeabilization with 0.3% Triton X-100 in PBS, cells were incubated with TUNEL reaction solution (In Situ Cell Death Detection Kit, TMR red, Roche, Switzerland). The mice were euthanized, and their hearts were isolated and cut into two transverse slices at the mid-level of the papillary muscles. The samples were fixed with Optimal Cutting Temperature Compound (Sakura Finetek Japan Co., Ltd., Japan) and then in 10% formalin. They were quenched with PBS containing 100 mM Tris-HCl and 0.1% Triton X-100, and were then permeabilized with 0.3% Triton X-100 in PBS. After permeabilization, they were incubated with TUNEL reaction solution. Following the application of a TrueView Autofluorescence Quenching Kit (Vector Laboratories, USA) to reduce cellular autofluorescence signal in cardiac tissue, the samples were incubated at room temperature. Nuclear staining was then performed with 1 µg/mL Hoechest 33258 (Dojindo, Japan) for 2 hours at 4℃. Fluorescence was observed with a fluorescence microscope (BZ-X810, Keyence, Japan). To determine the percentage of apoptotic cells, TUNEL-positive nuclei and TUNEL-negative cells were counted using Image J software, version 1.51 (U. S. National Institutes of Health, USA). Samples from at least three independent experiments were scored blindly.”

Line 185 on Page 6, Figure legends

“Figure 2. DOX-induced apoptosis was inhibited by CME in H9C2 cells.

H9C2 cells were pretreated with 1 mg/mL CME. Two hours later, the cells were treated with 1 µM DOX. Twenty-four hours after treatment, a TUNEL assay and nuclear staining were performed. (A) Representative images of TUNEL staining and nuclear staining with 1 µg/mL Hoechest 33258 of H9C2 cells. Arrows show TUNEL positive cells. (B) TUNEL positive ratio is defined as the number of TUNEL positive cells divided by the total number of cells. Values are presented as the mean ± SEM of three individual experiments.”

Line 218 on Page 7, Figure legends

“Figure 5. Survival rate that had been decreased by DOX was improved by CME.

Survival rate was determined for 12 days after intraperitoneal injection of 20 mg/kg DOX. Blue, vehicle (n = 10); red, DOX (n = 9); purple, 400 mg/kg CME + DOX (n = 10). *p < 0.01.”

Line 244 on Page 9, Figure legends

“Figure 7. DOX-induced apoptosis in mouse heart was inhibited by CME.

(A) Representative images of TUNEL staining and nuclear staining with 1 µg/mL Hoechest 33258 of mouse heart. Arrows show TUNEL positive cells. (B) TUNEL positive ratio is defined as the number of TUNEL positive cells divided by the total number of cells. Values are presented as the mean ± SEM of three individual experiments.”

Comment 1-3:

As for the western blotting, the overall results look quite convincing. Not only the blotting shapes look decent, but also, the quantifications of proteins in different treatments indicate the influence of the CME on the doxorubicin therapy. However, it is also important to perform Western Blotting in one cancer cell line to correlate the effects with the MTT data.

Response 1-3:

Thank you very much for your valuable suggestion. We have carried out additional experiments and found that 1 mg/mL CME did not inhibit DOX-induced upregulation of p53 activity in MDA-MB-231 cells. We have revised the Method, Results, Figure legends, and Discussion sections.

Line 94 on Page 4, Materials and Methods

“Primary cultured neonatal rat cardiomyocytes were isolated from 1- to 2-day-old SD rats as described previously [20, 49]. H9C2 cells, MDA-MB-231 cells, and H1299 cells were purchased from American Type Culture Collection (USA). HT29 cells were kindly provided by Prof. Akira Asai (University of Shizuoka, Japan). These cells were cultured in Dulbecco’s Modified Eagle’s Medium (Nacalai Tesque, Japan) with fetal bovine serum (FBS) and 1% penicil-lin-streptomycin-glutamine (Invitrogen, USA) in a humidified incubator at 37°C with 5% CO2. These cells were treated with 0.3 or 1 mg/mL CME for 2 h, followed by stimulation with 1 µM DOX. Primary cultured cardiomyocytes were incubated for 24h for the MTT assay. H9C2 cells were incubated for 24 h for the MTT assay and 12 h for protein extraction. MDA-MB-231 cells were incubated for 48 h for the MTT assay and 24 h for protein extraction. H1299 cells and HT29 cells were incubated for 48 h for the MTT assay.”

Line 121 on Page 4, Materials and Methods

“Protein extracts and nuclear extracts were obtained from H9C2 cells and MDA-MB-231 cells. Western blotting was performed as previously described [27-29]. For western blotting, anti-cleaved caspase-3 antibody, anti-cleaved caspase-9 antibody (Abcam, UK), anti-p53 antibody, phospho-p53 (Ser15) antibody (Cell Signaling Technology, USA), and anti α-tubulin monoclonal antibody (Fujifilm Wako Pure Chemical Corporation, Japan) were used as primary antibodies, and goat anti-rabbit IgG–HRP (MBL, Japan) and sheep anti-mouse IgG-HRP (GE Healthcare, USA) were used as secondary antibodies.”

Line 191 on Page 7, Results and Figure legends

“3.3. CME inhibited DOX-induced upregulation of p53, p-p53, cleaved caspase-3, and cleaved caspase-9       

 First, to investigate whether CME affects the expression levels of p53, phosphorylated p53 (p-p53), cleaved caspase-3, and cleaved caspase-9, H9C2 cells were pretreated with 1 mg/mL CME for 2 h and then incubated with 1 µM DOX for 12 h. One mg/mL CME significantly inhibited DOX-induced upregulation of p53 and p-p53 (Fig. 3A-C). Moreover, 1 mg/mL CME inhibited DOX-induced increases in cleaved caspase-3 and -9 activity (Fig. 3D-F). Next, to determine whether CME inhibits DOX-induced upregulation of p53, MDA-MB-231 cells were pretreated with 1 mg/mL CME for 2 h and then incubated with 1 µM DOX for 24 h. Western blotting demonstrated that 1 mg/mL CME did not affect DOX-induced increases in p53 upregulation (Fig. 3G). These results revealed that CME inhibited DOX-induced apoptosis by inhibiting p53 and cleaved casapase-3 activity.

Figure 3. DOX-induced protein expression of cellular apoptosis markers was inhibited by CME.

H9C2 cells were pretreated with 1mg/mL CME for 2 h, and then cardiac cytotoxicity was induced with 1 µM DOX for 12 h. MDA-MB-231 cells were pretreated with 1mg/mL CME for 2 h, and then cardiac cytotoxicity was induced with 1 µM DOX for 24 h. (A, D) Representative images of WB in H9C2 cells. (B, C, E, F) Expression levels of p53 (B), phosph-p53 (C), cleaved caspase-3 (E), and cleaved caspase-9 (F) were calculated as ratios relative to β-actin. (G) Representative images of WB in MDA-MB-231 cells. Values are presented as the mean ± SEM of three individual experiments.”

Line 254 on Page 9, Discussion

“CME also suppressed in vitro the activation of caspase-3 and -9, which regulate DOX-induced apoptosis, as well as that of the tumor suppressor gene p53, which is upstream of these caspases in H9C2 cells. However, CME did not affect DOX-induced upregulation of p53 in MDA-MB-231 cells. DOX is known to cause DNA damage by the interaction of three factors: ROS production and a subsequent increase in oxidative stress, mitochondrial metabolic dysfunction, and the interaction of damaged DNA with TOP2β. This DNA damage then activates p53 [34-36].  CME has been reported to possess mainly anti-inflammatory and anti-tumor activities [37-39]. The results of the present study show that CME inhibited the activation of p53. This suggests that CME inhibits the activation of p53 by affecting oxidative stress, mitochondrial dysfunction, and the interaction of damaged DNA with TOP2β in cardiomyocytes.”

Comment 1-4:

Overall, I like the manuscript very much, I think that the manuscript is a good fit for the journal and will be well cited.

I have one major comment that must be addressed. The authours mention that CME is commercially available. In that case they should give the catalogue number, purity and other details. If it is not available online, the authours should provide the data about the composition of the CME – HPLC, types of fractions, purity, etc. Without this data, it is close to impossible to reproduce these results.

Response 1-4:

Thank you very much for your favorable comment and kind advice. In response, we have modified Method section.

Line 94 on Page 4, Materials and Methods

“CME was obtained from UNIAL Co., Ltd. (Japan). It was extracted with hot water and dried into a powder. The extract is a standardized product that includes chlorogenic acid (420 mg/100 g), delphinidin (690 mg/100 g), luteolin (32 mg/100 g), and total polyphenol (4.75 g/100 g) derived from Chrysanthemum morifolium Ramat. CME is available through the website of UNIAL Co., Ltd. (http://www.unial.info/materials/beauty.html). Doxorubicin hydrochloride was purchased from MedChemExpress (China) and stored at -20℃ until use.”

Reviewer 2 Report

The work by Ono M. et al. aims at evaluating the cardioprotective effect of Chrysanthemum morifolium extract (CME) during doxorubicin (DOX) treatment. CME has been reported to possess various beneficial properties, including antioxidant and anti-inflammatory activities, promoting this molecule as a promising candidate drug in limiting DOX-induced cardiotoxicity. The in vitro experiments revealed that the CME pre-treatment counteracted the DOX-dependent activation of the apoptotic pathway, reducing the expression of p-53 and cleaved caspase-3,9 in H9C2 cells. These findings were corroborated in a mouse model of DOX-induced cardiotoxicity, showing that CME administration prevented cardiac dysfunction and reduced the apoptosis rate in the cardiac tissue of mice treated with a single injection of DOX. However, several issues negatively affect the quality of the manuscript.

 MAJOR POINTS

-Despite the H9C2 cells are often used for in vitro investigation, the generated data should be corroborated in a more representative model, such as neonatal cardiomyocytes.

-In all the experiments, the effect of CME administration alone is not shown. Did the authors have any data about the effect of CME as a single agent? Importantly, a recent paper described CME-related cytotoxicity activity against different cell types, including breast cancer cells at a concentration of about 300 µg/mL (34083561). Concerning that this dose is three times low compared to the one used in this work, should the author consider evaluating the effect of CME alone on another type of breast cancer cell lines, such as MCF-7?

-The choice of the mouse model used for the in vivo investigation is questionable. The dose of DOX administered to the animals (20 mg/Kg) is not well tolerated by the animals, as confirmed by the increased mortality rate in Fig.4. Additionally, this chemotherapeutic regimen does not recapitulate the chronic treatment underwent by cancer patients. A different model is recommended to validate the in vivo data obtained by the authors.

 -The results shown in Fig.6 are not convincing. The image does not support the protective effect of CME against DOX and the percentage of positive cells revealed by the Tunel assay is not relevant.

-The cardioprotective effect of CME should be confirmed through a detailed histological analysis, including the measurement of cardiac cell area and collagen deposition, associated with the evaluation of conventional markers of cardiac stress, such as ANP and BNP.

-It would be important to use human stem cells derived cardiomyocytes and endothelial cells to check whether CME would have its protective effect also in human cells in in vitro settings.

Author Response

Reviewer 2

Comments and Suggestions for Authors

The work by Ono M. et al. aims at evaluating the cardioprotective effect of Chrysanthemum morifolium extract (CME) during doxorubicin (DOX) treatment. CME has been reported to possess various beneficial properties, including antioxidant and anti-inflammatory activities, promoting this molecule as a promising candidate drug in limiting DOX-induced cardiotoxicity. The in vitro experiments revealed that the CME pre-treatment counteracted the DOX-dependent activation of the apoptotic pathway, reducing the expression of p-53 and cleaved caspase-3,9 in H9C2 cells. These findings were corroborated in a mouse model of DOX-induced cardiotoxicity, showing that CME administration prevented cardiac dysfunction and reduced the apoptosis rate in the cardiac tissue of mice treated with a single injection of DOX. However, several issues negatively affect the quality of the manuscript.

Comment 2-1:

Despite the H9C2 cells are often used for in vitro investigation, the generated data should be corroborated in a more representative model, such as neonatal cardiomyocytes.

Response 2-1:

Thank you very much for your valuable comment. In response, we have carried out an MTT assay with primary cultured cardiomyocytes and found that CME also inhibited DOX-induced cardiac cytotoxicity. We have revised the Method, Results, and Discussion sections.

Line 98 on Page 4, Materials and Methods

“Primary cultured neonatal rat cardiomyocytes were isolated from 1- to 2-day-old SD rats as described previously [20, 49]. H9C2 cells, MDA-MB-231 cells, and H1299 cells were purchased from American Type Culture Collection (USA). HT29 cells were kindly provided by Prof. Akira Asai (University of Shizuoka, Japan). These cells were cultured in Dulbecco’s Modified Eagle’s Medium (Nacalai Tesque, Japan) with fetal bovine serum (FBS) and 1% penicillin-streptomycin-glutamine (Invitrogen, USA) in a humidified incubator at 37°C with 5% CO2. These cells were treated with 0.3 or 1 mg/mL CME for 2 h, followed by stimulation with 1 µM DOX. H9C2 cells were incubated for 24 h for the MTT assay and 12 h for protein extraction. Primary cultured cardiomyocytes were incubated for 24h for the MTT assay. MDA-MB-231 cells were incubated for 48 h for the MTT assay and 24 h for protein extraction. H1299 cells and HT29 cells were incubated for 48 h for the MTT assay.”

Line 107 on Page 4, Materials and Methods

“H9C2 cells, primary cultured cardiomyocytes, MDA-MB-231 cells, H1299 cells, and HT29 cells were washed with serum free medium and added to Counting Kit-8 medium (Dojindo, Japan) for 1 h. After incubation, measurement of absorbance was performed using a Wallac 1420 Arvo Sx multilabel counter (Perkin Elmer, USA). The relative percentage of cell survival was calculated by dividing the absorbance of the treated cells by that of the control in each experiment.”

Line 163 on Page 5, Results

3.1. CME inhibited DOX-induced cytotoxicity in H9C2 cells and primary cultured cardiomyocytes.

The effect of CME on DOX-induced cytotoxicity was examined using an MTT assay with H9C2 cells and primary cultured cardiomyocytes (Fig. 1A, B). DOX reduced the viability of these cells to 29% and 28%, respectively, but 1 mg/mL CME inhibited DOX-induced cell cytotoxicity, increasing cell viability to 75% and 79%, respectively. An MTT assay also revealed that 1 mg/mL CME did not induce cytotoxicity in H9C2 cells (Fig. 1C). Next, to determine whether CME reduces the anti-tumor activity of DOX, MDA-MB-231 human breast cancer cells, H1299 cells human non-small cell lung carcinoma cells and HT29 human colon cancer cells were treated with CME and DOX. Cell viability was decreased by DOX, but 1 mg/mL CME did not affect viability (Fig. 1D-F). These results indicate that CME inhibited the cytotoxicity induced by DOX without reducing its antitumor activity.”

Line 172 on Page 5, Figure legends

Figure 1. DOX-induced cytotoxicity was inhibited by CME in H9C2 cells and primary cultured cardiomyocytes.

(A) H9C2 cells were pretreated with 0.3 or 1 mg/mL CME. After 2 h, the cells were treated with 1 µM DOX for 24 h. Cell viability was investigated by MTT assay. (B, C) H9C2 cells and primary cultured cardiomyocytes were pretreated with 1 mg/mL CME for 2 h and then treated with 1 µM DOX for 24 h. (D-F) MDA-MB-231 cells (D), H1299 cells (E), and HT29 cells (F) were pretreated with 1 mg/mL CME. These cells were stimulated with 1 µM DOX for 48 h. Cell viability was measured by MTT assay. Values are presented as the mean ± SEM of three individual experiments.”

Line 250 on Page 10, Discussion

“This study found that CME suppressed DOX-induced cytotoxicity in H9C2 cells and primary cultured myocytes without inhibiting the anti-tumor effect of DOX in MDA-MB-231 cells, H1299 cells and HT29 cells. This suggests that CME suppresses DOX-induced cardiotoxicity without inhibiting the anti-tumor effect of DOX”

Line 287 on Page 9, Institutional Review Board Statement

“Institutional Review Board Statement: All animal experiments complied with the guidelines on animal experiments of the University of Shizuoka and the National Hospital Organization Kyoto Medical Center and were performed in accordance with protocols approved by the University of Shizuoka Ethics Committee (numbers 176278 and 186353) and the National Hospital Organization Kyoto Medical Center Ethics Committee (number 30-29-1).”

Comment 2-2:

In all the experiments, the effect of CME administration alone is not shown. Did the authors have any data about the effect of CME as a single agent? Importantly, a recent paper described CME-related cytotoxicity activity against different cell types, including breast cancer cells at a concentration of about 300 µg/mL (34083561). Concerning that this dose is three times low compared to the one used in this work, should the author consider evaluating the effect of CME alone on another type of breast cancer cell lines, such as MCF-7?

Response 2-2:

                 Thank you very much for your advice. In response, we have carried out an MTT assay to evaluate the effect of CME alone on MDA-MB-231 cells, H1299 human non-small cell lung carcinoma cells, and HT29 human colon cancer cells. We have found that 1 mg/mL CME did not have an anti-tumor effect against MDA-MB-231, H1299, or HT29 cells. We have modified the Method, Results, and Discussion sections.

The CME used in this study was provided from Unial Co., Ltd. This extract is a standardized product that includes chlorogenic acid (420 mg/100 g), delphinidin (690 mg/100 g), luteolin (32 mg/100 g), and total polyphenol (4.75 g/100 g). CME at 1 mg/mL contains 12 µM chlorogenic acid, 23 µM delphinidin, and 1.1 µM luteolin. Previous reports have shown that chlorogenic acid (up to 200 µM), delphinidin (up to 100 µM), luteolin (up to 20 µM) had no cytotoxicity on H9C2 cells (Rf.1-3). Chlorogenic acid at 200 µM, delphinidin at 100 µM, and luteolin at 5 µM attenuated Dox-induced cytotoxicity in H9C2 cells (Rf.1,3,4). While chlorogenic acid and delphinidin did not affect Dox-induced cytotoxicity in MG-63 human osteosarcoma cells or MCF-7 human breast cancer cells, respectively, luteolin at 10 µM attenuated Dox-induced cytotoxicity in MCF-7 cells (Rf.5). Thus, the concentrations of these compounds in CME in the present study were low compared to those in past studies. The additional or synergistic effects of these compounds may contribute to the suppression of DOX-induced cytotoxicity in H9C2 cells and in vivo. Further studies are needed to clarify this issue.

  • Salzillo A, Ragone A, Spina A, Naviglio S, Sapio L. Chlorogenic Acid Enhances Doxorubicin-Mediated Cytotoxic Effect in Osteosarcoma Cells. Int J Mol Sci. 2021 Aug 10;22(16):8586.
  • Chang H, Li C, Huo K, Wang Q, Lu L, Zhang Q, Wang Y, Wang W. Luteolin Prevents H2O2-Induced Apoptosis in H9C2 Cells through Modulating Akt-P53/Mdm2 Signaling Pathway. Biomed Res Int. 2016;2016:5125836.
  • Choi EH, Chang HJ, Cho JY, Chun HS. Cytoprotective effect of anthocyanins against doxorubicin-induced toxicity in H9c2 cardiomyocytes in relation to their antioxidant activities. Food Chem Toxicol. 2007 Oct;45(10):1873-81.
  • Shi Y, Li F, Shen M, Sun C, Hao W, Wu C, Xie Y, Zhang S, Gao H, Yang J, Zhou Z, Gao D, Qin Y, Han X, Liu S. Luteolin Prevents Cardiac Dysfunction and Improves the Chemotherapeutic Efficacy of Doxorubicin in Breast Cancer. Front Cardiovasc Med. 2021 Oct 13;8:750186.
  • Sato Y, Sasaki N, Saito M, Endo N, Kugawa F, Ueno A. Luteolin attenuates doxorubicin-induced cytotoxicity to MCF-7 human breast cancer cells. Biol Pharm Bull. 2015;38(5):703-9.

Line 94 on Page 4, Materials and Methods

“CME was obtained from UNIAL Co., Ltd. (Japan). It was extracted with hot water and dried into a powder. The extract is a standardized product that includes chlorogenic acid (420 mg/100 g), delphinidin (690 mg/100 g), luteolin (32 mg/100 g), and total polyphenol (4.75 g/100 g) derived from Chrysanthemum morifolium Ramat. CME is available through the website of UNIAL Co., Ltd. (http://www.unial.info/materials/beauty.html). Doxorubicin hydrochloride was purchased from MedChemExpress (China) and stored at -20℃ until use.”

Line 98 on Page 4, Materials and Methods

“Primary cultured neonatal rat cardiomyocytes were isolated from 1- to 2-day-old SD rats as described previously [20, 49]. H9C2 cells, MDA-MB-231 cells, and H1299 cells were purchased from American Type Culture Collection (USA). HT29 cells were kindly provided by Prof. Akira Asai (University of Shizuoka, Japan). These cells were cultured in Dulbecco’s Modified Eagle’s Medium (Nacalai Tesque, Japan) with fetal bovine serum (FBS) and 1% penicillin-streptomycin-glutamine (Invitrogen, USA) in a humidified incubator at 37°C with 5% CO2. These cells were treated with 0.3 or 1 mg/mL CME for 2 h, followed by stimulation with 1 µM DOX. Primary cultured cardiomyocytes were incubated for 24h for the MTT assay. H9C2 cells were incubated for 24 h for the MTT assay and 12 h for protein extraction. MDA-MB-231 cells were incubated for 48 h for the MTT assay and 24 h for protein extraction. H1299 cells, and HT29 cells were incubated for 48 h for the MTT assay.”

Line 107 on Page 4, Materials and Methods

“Primary cultured cardiomyocytes, H9C2 cells, MDA-MB-231 cells, H1299 cells, and HT29 cells were washed with serum free medium and added to Counting Kit-8 medium (Dojindo, Japan) for 1 h. After incubation, measurement of absorbance was performed using a Wallac 1420 Arvo Sx multilabel counter (Perkin Elmer, USA). The relative percentage of cell survival was calculated by dividing the absorbance of the treated cells by that of the control in each experiment.”

Line 163 on Page 5, Results

3.1. CME inhibited DOX-induced cytotoxicity in H9C2 cells and primary cultured cardiomyocytes.

The effect of CME on DOX-induced cytotoxicity was examined using an MTT assay with H9C2 cells and primary cultured cardiomyocytes (Fig. 1A, B). DOX reduced the viability of these cells to 29% and 28%, respectively, but 1 mg/mL CME inhibited DOX-induced cell cytotoxicity, increasing cell viability to 75% and 79%, respectively. An MTT assay also revealed that 1 mg/mL CME did not induce cytotoxicity in H9C2 cells (Fig. 1C). Next, to determine whether CME reduces the anti-tumor activity of DOX, MDA-MB-231 human breast cancer cells, H1299 cells human non-small cell lung carcinoma cells and HT29 human colon cancer cells were treated with CME and DOX. Cell viability was decreased by DOX, but 1 mg/mL CME did not affect viability (Fig. 1D-F). These results indicate that CME inhibited the cytotoxicity induced by DOX without reducing its antitumor activity.”

Line 172 on Page 5, Figure legends

Figure 1. DOX-induced cytotoxicity was inhibited by CME in H9C2 cells and primary cultured cardiomyocytes.

(A) H9C2 cells were pretreated with 0.3 or 1 mg/mL CME. After 2 h, the cells were treated with 1 µM DOX for 24 h. Cell viability was investigated by MTT assay. (B, C) H9C2 cells and primary cultured cardiomyocytes were pretreated with 1 mg/mL CME for 2 h and then treated with 1 µM DOX for 24 h. (D-F) MDA-MB-231 cells (D), H1299 cells (E), and HT29 cells (F) were pretreated with 1 mg/mL CME. These cells were stimulated with 1 µM DOX for 48 h. Cell viability was measured by MTT assay. Values are presented as the mean ± SEM of three individual experiments.”

Line 250 on Page 10, Discussion

“This study found that CME suppressed DOX-induced cytotoxicity in H9C2 cells and primary cultured myocytes without inhibiting the anti-tumor effect of DOX in MDA-MB-231 cells, H1299 cells and HT29 cells. This suggests that CME suppresses DOX-induced cardiotoxicity without inhibiting the anti-tumor effect of DOX”

Line 285 on Page 10, Discussion

“A previous study evaluating the safety of CME showed that no toxicity was observed even after daily oral administration of 1280 mg/kg to SD rats for 26 weeks [48]. As Chrysanthemum morifolium has been eaten since ancient times, it is likely to be safe. However, further clinical safety studies on CME should still be carried out.

This study has several limitations. CME is known to contain large amounts of delphinidin, luteolin, and chlorogenic acid; however, in this study we were unable to identify which compounds contributed to the beneficial effect of CME on DOX-induced cardiotoxicity. Second, to determine the effect of CME, this study focused only on the acute cardiotoxicity induced by a single high dose of DOX. Therefore, based on this study, it is not possible to predict the protective effect of CME on chronic cardiomyopathy induced by low doses of DOX. Recently, human pluripotent stem cell-derived cardiomyocytes and endothelial cells have emerged as useful tools for analyzing cardiotoxicity in physiologically relevant human cells [51]. As this study confirmed the safety of CME only in primary cultured cardiomyocytes and H9C2 cells, future studies are needed to clarify the safety of the compound in human pluripotent stem cell-derived cardiomyocytes and endothelial cells in order to predict potential adverse effects of DOX in the clinical setting.”

Line 287 on Page 9, Institutional Review Board Statement

“Institutional Review Board Statement: All animal experiments complied with the guidelines on animal experiments of the University of Shizuoka and the National Hospital Organization Kyoto Medical Center and were performed in accordance with protocols approved by the University of Shizuoka Ethics Committee (numbers 176278 and 186353) and the National Hospital Organization Kyoto Medical Center Ethics Committee (number 30-29-1).”

Comment 2-3:

The choice of the mouse model used for the in vivo investigation is questionable. The dose of DOX administered to the animals (20 mg/Kg) is not well tolerated by the animals, as confirmed by the increased mortality rate in Fig.4. Additionally, this chemotherapeutic regimen does not recapitulate the chronic treatment underwent by cancer patients. A different model is recommended to validate the in vivo data obtained by the authors.

Response 2-3:

                 Thank you for your suggestion. Our study focuses on acute cardiotoxicity with a single high dose of DOX to determine the effect of CME. Future studies are needed to clarify the protective effect of CME on chronic cardiomyopathy induced by low doses of DOX. This limitation is described in the revised manuscript.

Line 285 on Page 10, Discussion

“A previous study evaluating the safety of CME showed that no toxicity was observed even after daily oral administration of 1280 mg/kg to SD rats for 26 weeks [48]. As Chrysanthemum morifolium has been eaten since ancient times, it is likely to be safe. However, further clinical safety studies on CME should still be carried out.

This study has several limitations. CME is known to contain large amounts of delphinidin, luteolin, and chlorogenic acid; however, in this study we were unable to identify which compounds contributed to the beneficial effect of CME on DOX-induced cardiotoxicity. Second, to determine the effect of CME, this study focused only on the acute cardiotoxicity induced by a single high dose of DOX. Therefore, based on this study, it is not possible to predict the protective effect of CME on chronic cardiomyopathy induced by low doses of DOX. Recently, human pluripotent stem cell-derived cardiomyocytes and endothelial cells have emerged as useful tools for analyzing cardiotoxicity in physiologically relevant human cells [51]. As this study confirmed the safety of CME only in primary cultured cardiomyocytes and H9C2 cells, future studies are needed to clarify the safety of the compound in human pluripotent stem cell-derived cardiomyocytes and endothelial cells in order to predict potential adverse effects of DOX in the clinical setting.”

Comment 2-4:

The results shown in Fig.6 are not convincing. The image does not support the protective effect of CME against DOX and the percentage of positive cells revealed by the Tunel assay is not relevant.

Response 2-4:

                 Thank you for your comment. Researchers vary in their views on the number of TUNEL-positive cells required to induce cardiomyopathy in the heart. Low levels of cardiac myocyte apoptosis are component in the pathogenesis of heart failure [Rf.7,8] and Wencker et al. reported that low levels of myocyte apoptosis (23 myocytes per 105 nuclei) were sufficient to cause lethal dilated cardiomyopathy in mice with cardiac-specific expression of caspase-8 fusion protein (Rf.6). The present study showed that 26 myocytes per 105 (2.6×10-2%) nuclei were detected at one week after DOX administration. We consider this ratio sufficient to induce cardiac dysfunction in vivo. We have modified the Discussion section.

  • Wencker D, Chandra M, Nguyen K, Miao W, Garantziotis S, Factor SM, Shirani J, Armstrong RC, Kitsis RN. A mechanistic role for cardiac myocyte apoptosis in heart failure. J Clin Invest. 2003 May;111(10):1497-504.
  • Yaoita H, Ogawa K, Maehara K, Maruyama Y. Attenuation of ischemia/reperfusion injury in rats by a caspase inhibitor. Circulation.1998; 97:276–281.
  • Latif N. Khan M.A. Birks E. O'Farrell A. Westbrook J. Dunn M.J. et al. Upregulation of the Bcl-2 family of proteins in end stage heart failure J Am Coll Cardiol 2000 35 1769 1777

Line 264 on Page 9, Discussion

“               To investigate whether CME inhibits DOX-induced cardiotoxicity in vivo, we used DOX-induced heart failure model mice. Echocardiography showed that CME suppressed cardiac dysfunction due to DOX, and TUNEL staining revealed that CME suppressed DOX-induced apoptosis in vivo. Wencker et al. reported that low levels of myocyte apoptosis (23 myocytes per 105 nuclei) were sufficient to cause lethal dilated cardiomyopathy in mice with cardiac-specific expression of caspase-8 fusion protein [50]; as the present study detected 26 myocytes per 105 (2.6×10-2%) nuclei at one week after DOX administration, this ratio can be considered sufficient to have induced cardiac dysfunction in vivo. These results indicate that CME improves DOX-induced cardiotoxicity by suppressing apoptosis, which is a major cause of the progression of heart failure.”

Comment 2-5:

The cardioprotective effect of CME should be confirmed through a detailed histological analysis, including the measurement of cardiac cell area and collagen deposition, associated with the evaluation of conventional markers of cardiac stress, such as ANP and BNP.

Response 2-5:

                 Thank you for your comment. In response to it, we have measured myocardial cell surface area, perivascular fibrosis area, and cardiac stress markers, such as ANF and BNP, and found that the administration of DOX did not change these parameters. Previous studies have also shown that the administration of DOX for 1-2 weeks did not change cardiac morphology, collagen, or fibrosis in heart tissue (Rf. 7, 8). Thus, if such changes do in fact occur, they may not occur within the short administration time that DOX was given in this study. However, in the experiments carried out prior to our initial submission, we also confirmed that DOX reduced survival rate and HW/TL ratio, induced cardiac dysfunction, and increased the number of TUNEL-positive cells in cardiac tissue, all of which were significantly improved by CME treatment. Therefore, we think that CME possesses a cardioprotective effect against DOX-induced cardiotoxicity in vivo. We have added the results of these earlier experiments as Supplemental Figure 1 and have described some related limitations of the study in the revised manuscript.

  • Zilinyi R, Czompa A, Czegledi A, Gajtko A, Pituk D, Lekli I, Tosaki A. The Cardioprotective Effect of Metformin in Doxorubicin-Induced Cardiotoxicity: The Role of Autophagy. Molecules. 2018 May 15;23(5):1184.
  • Jun-Jie Guo, Lei-Lei Ma, Hong-Tao Shi, Jian-Bing Zhu, Jian Wu, Zhi-Wen Ding, Yi An, Yun-Zeng Zou, Jun-Bo Ge. Alginate Oligosaccharide Prevents Acute Doxorubicin Cardiotoxicity by Suppressing Oxidative Stress and Endoplasmic Reticulum-Mediated Apoptosis. Mar Drugs. 2016 Dec; 14(12): 231.

Line 237 on Page 8, Results

“3.6. CME suppressed DOX-induced cardiac apoptosis in mice.

                 To determine the protective effect of CME against DOX-induced cardiac damage, a histological analysis was carried out. Representative myocardial cross-sectional images stained with hemotoxin/eosin and picrosirius red are shown in Supplemental Figure 1. There were no significant differences in cardiac morphology, collagen, or fibrosis in the heart tissue between the DOX-treated group and the control group. TUNEL staining was performed to investigate the effect of CME on DOX-induced cardiac apoptosis in vivo (Fig. 6A). Apoptosis in the heart was increased by DOX, and this increase was suppressed by CME (Fig. 6B). These results reveal that CME improved DOX-induced cardiac dysfunction by inhibiting DOX-induced apoptosis.”

Line 285 on Page 10, Discussion

“A previous study evaluating the safety of CME showed that no toxicity was observed even after daily oral administration of 1280 mg/kg to SD rats for 26 weeks [48]. As Chrysanthemum morifolium has been eaten since ancient times, it is likely to be safe. However, further clinical safety studies on CME should still be carried out.

This study has several limitations. CME is known to contain large amounts of delphinidin, luteolin, and chlorogenic acid; however, in this study we were unable to identify which compounds contributed to the beneficial effect of CME on DOX-induced cardiotoxicity. Second, to determine the effect of CME, this study focused only on the acute cardiotoxicity induced by a single high dose of DOX. Therefore, based on this study, it is not possible to predict the protective effect of CME on chronic cardiomyopathy induced by low doses of DOX. Recently, human pluripotent stem cell-derived cardiomyocytes and endothelial cells have emerged as useful tools for analyzing cardiotoxicity in physiologically relevant human cells [51]. As this study confirmed the safety of CME only in primary cultured cardiomyocytes and H9C2 cells, future studies are needed to clarify the safety of the compound in human pluripotent stem cell-derived cardiomyocytes and endothelial cells in order to predict potential adverse effects of DOX in the clinical setting.”

Comment 2-6:

It would be important to use human stem cells derived cardiomyocytes and endothelial cells to check whether CME would have its protective effect also in human cells in in vitro settings.

Response 2-6:

                 Thank you for your valuable suggestion. Recently, human pluripotent stem cell-derived cardiomyocytes and endothelial cells have emerged as a very useful tool to analyze cardiotoxicity in highly physiologically relevant human cells (Rf.9). In this study, we confirmed that the safety of CME only on primary cultured cardiomyocytes and H9C2 cells. To reduce the adverse effects of DOX in the clinical setting, we should clarify the safety of CME using human pluripotent stem cell-derived cardiomyocytes and endothelial cells in a future study. We have added discussion about this to the revised manuscript.

Rf.9  Ni X, Yang ZZ, Ye LQ, Han XL, Zhao DD, Ding FY, Ding N, Wu HC, Yu M, Xu GY, Zhao ZA, Lei W, Hu SJ. Establishment of an in vitro safety assessment model for lipid-lowering drugs using same-origin human pluripotent stem cell-derived cardiomyocytes and endothelial cells. Acta Pharmacol Sin. 2022 Jan;43(1):240-250.

Line 285 on Page 10, Discussion

“A previous study evaluating the safety of CME showed that no toxicity was observed even after daily oral administration of 1280 mg/kg to SD rats for 26 weeks [48]. As Chrysanthemum morifolium has been eaten since ancient times, it is likely to be safe. However, further clinical safety studies on CME should still be carried out.

This study has several limitations. CME is known to contain large amounts of delphinidin, luteolin, and chlorogenic acid; however, in this study we were unable to identify which compounds contributed to the beneficial effect of CME on DOX-induced cardiotoxicity. Second, to determine the effect of CME, this study focused only on the acute cardiotoxicity induced by a single high dose of DOX. Therefore, based on this study, it is not possible to predict the protective effect of CME on chronic cardiomyopathy induced by low doses of DOX. Recently, human pluripotent stem cell-derived cardiomyocytes and endothelial cells have emerged as useful tools for analyzing cardiotoxicity in physiologically relevant human cells [51]. As this study confirmed the safety of CME only in primary cultured cardiomyocytes and H9C2 cells, future studies are needed to clarify the safety of the compound in human pluripotent stem cell-derived cardiomyocytes and endothelial cells in order to predict potential adverse effects of DOX in the clinical setting.”

Round 2

Reviewer 2 Report

I thank the authors for exhaustively answering all the points. Overall, the quality of the manuscript is significantly improved. I have no further comments or suggestions.